# UNDERSTANDING CONTEXTUAL RECALL IN TRANSFORMERS: HOW FINETUNING ENABLES IN-CONTEXT REASONING OVER PRETRAINING KNOWLEDGE

## ABSTRACT

Transformer-based language models excel at in-context learning (ICL), where they can adapt to new tasks based on contextual examples, without parameter updates. In a specific form of ICL, which we refer to as *contextual recall*, models pretrained on open-ended text leverage pairwise examples to recall specific facts in novel prompt formats. We investigate whether contextual recall emerges from pretraining alone, what finetuning is required, and what mechanisms drive the necessary representations. For this, we introduce a controlled synthetic framework where pretraining sequences consist of subject-grammar-attribute tuples, with attribute types tied to grammar statistics. We demonstrate that while such pretraining successfully yields factual knowledge, it is insufficient for contextual recall: models fail to implicitly infer attribute types when the grammar statistics are removed in ICL prompts. However, we show that finetuning on tasks requiring implicit inference, distinct from the ICL evaluation, using a subset of subjects, triggers the emergence of contextual recall across all subjects. This transition is accompanied by the formation of low-dimensional latent encodings of the shared attribute type. For mechanistic insight, we derive a construction for an attention-only transformer that replicates the transition from factual to contextual recall, corroborated by empirical validation.

## 1 INTRODUCTION

Transformer-based large language models (LLMs) exhibit remarkable abilities to extrapolate far beyond tasks seen during training. A notable instance of this extrapolation is in-context learning (ICL) (Brown et al., 2020), where models can adapt to new tasks based on contextual examples, without parameter updates.

In this paper, we investigate a specific form of ICL, which we refer to as *contextual recall*. Here, a model trained on open-ended text acquires factual knowledge and is later able to recall specific facts when prompted with example pairs in an unseen format. To illustrate, pretraining data might include descriptions of various landmarks, from which the model learns multiple attributes for each—such as the country where it is located, the year it was built, or its architectural style. At test time, the model receives a prompt of the form `[Niagara Falls, Canada. Colosseum, Italy. Parthenon, ]` and must generate `[Greece]`. The prompt contains no explicit indication that the relevant attribute is "country" rather than, say, "year built"; the model must infer the attribute type from the in-context examples and recall the corresponding fact for the query subject.

The ability to succeed at contextual recall therefore requires both the acquisition of factual knowledge and adaptability to novel prompt formats, since the model may never have seen these facts presented as implicit subject-attribute pairs during pretraining (see Appendix A for a detailed discussion on related work on understanding ICL and factual recall using controlled, synthetic settings). In this work, we investigate the origins of this capability:

*Does the ability to do contextual recall emerge naturally from pretraining, or does it necessitate specific finetuning? Furthermore, what mechanisms within the Transformer architecture enable the emergence of this ability?*

**Contributions.** We summarize our contributions below.

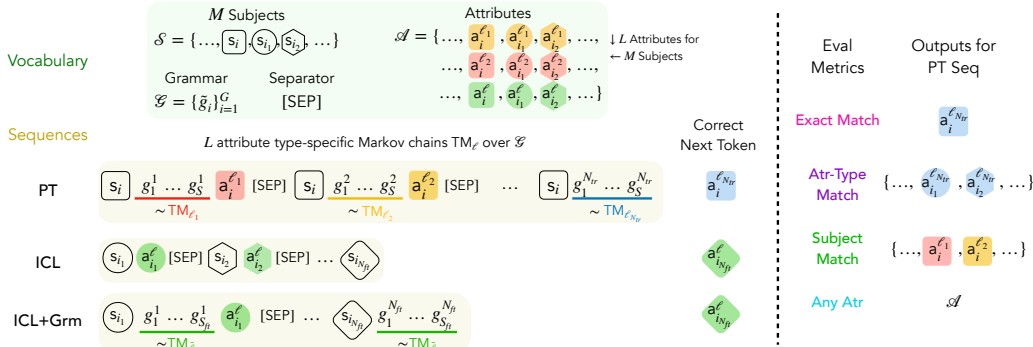

Figure 1: Illustration of the **data generation process** and the **evaluation metrics**. The vocabulary consists of $M$ subjects with $L$ attributes each (where the attribute tokens may be shared across the subjects), $G$ grammar tokens and a separator token . We sample $L$ Markov chains, one for each attribute type. We consider three types of sequences (see Appendix B for details): PT sequences contain subject-grammar-attribute tuples, with attribute type information encoded in the grammar sequence statistics, for one subject, ICL sequences contain pairwise subject-attribute examples for a shared attribute type across subjects, but no grammar, and ICL+Grm sequences are analogous to ICL sequences, but contain subject-grammar-attribute tuples. We evaluate whether the model's prediction (on PT sequences) matches the ground truth attribute (*Exact Match*), any attribute of the same type as the last subsequence (*Attribute-type Match*), any attribute that belongs to the same subject as the sequence (*Subject Match*), or any attribute token (*Any Attribute*).

First, we introduce a controlled synthetic framework to study the emergence of contextual recall in transformers. The pretraining (PT) sequences, used to instill factual knowledge in the model, contain multiple attributes of a single subject, interspersed with grammar tokens that encode information about each attribute type. The ICL sequences, used to evaluate contextual recall ability, consist of subject-attribute pairs sharing a common attribute type across different subjects, without any grammar. See Fig. 1 for an illustration of the data generation process and Appendix B for a detailed description.

In Section 2, we show that transformers trained with PT sequences succeed on factual recall, but fail to generalize to ICL sequences, which require the model to implicitly infer the attribute type from the in-context examples. However, we find that finetuning the model on sequences that are distinct from ICL sequences, but require implicit inference, using a subset of subjects, enables out-of-distribution generalization on ICL sequences on the held-out subjects (Section 3). We also probe the effect of some key dataset parameters on the model's performance on PT and ICL sequences in Appendix D.1.

In Section 4, we analyze model representations and find that finetuning induces the formation of low-dimensional encodings of the shared attribute type based on the in-context examples. These representations become more disentangled as the number of in-context demonstrations increases.

Finally, in Appendix C, for mechanistic insight, we consider a simpler synthetic task and present constructions for an attention-only model that succeeds on factual recall after pretraining and contextual recall after finetuning, and corroborate them with empirical validation.

## 2 PRETRAINING ON PT SEQUENCES

We train a two-layer, single-head, decoder-only transformer on PT sequences to minimize the standard next-token prediction objective. We use an online training setup: at each iteration, we generate a fresh batch of PT sequences using the process described in App. B. Unless stated otherwise, we fix $M = 256, L = 8, S = 80$. We evaluate the model on two distinct held-out sets: i) PT sequences, to evaluate factual recall, and ii) ICL sequences, to evaluate contextual recall capabilities. For both, we measure the accuracy of predicting the final token given the preceding context, using four metrics, as illustrated in Fig. 1. See App. D for further details.

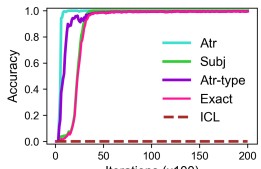

Figure 2: Transformer trained on PT sequences performs well on PT sequences, but does not generalize to ICL sequences.

Fig. 2 illustrates the performance as pretraining progresses. On the `PT` sequences (solid lines), we observe a stage-wise learning process: the model first learns to predict the attribute type, and then the exact attribute token. Crucially, however, we find that high performance on `PT` sequences does not transfer to `ICL` sequences, with the model achieving near-zero accuracy on `ICL` sequences. This shows that despite learning the factual associations, the model relies on *explicit* grammar statistics for retrieval and cannot *implicitly* infer the attribute type from in-context examples alone.

> **Finding 1**: Pretraining on `PT` sequences does *not* suffice for good `ICL` performance.

## 3 FINETUNING EXPERIMENTS

Since pretraining on `PT` sequences alone does not suffice for good `ICL` performance, we investigate whether finetuning can bridge this gap. Specifically, we ask: can a model originally trained to rely on explicit grammar-based cues be adapted to perform implicit inference from in-context examples? To answer this, we finetune the pretrained model using the standard next-token prediction objective on a new data distribution (detailed below). Crucially, to test for generalization, we finetune on only a subset of subjects, reserving the remaining subjects as a held-out set. We then evaluate the model on `ICL` sequences (as defined in Appendix B) where the query subject belongs to this held-out set. We set the number of demonstrations $N_{\mathrm{ft}} = 16$, and use $50\%$ of the total subjects for finetuning.

**FT on `ICL` using a subset of subjects.** We first consider the most direct approach: finetuning the pretrained model directly on `ICL` sequences. This serves as an upper bound, since the finetuning and evaluation formats are identical, *i.e.*, there is no distribution shift in prompt format. Fig. 3 shows performance on `ICL` sequences with seen and held-out query subjects. As finetuning progresses, the model successfully learns to perform contextual recall, generalizing to held-out subjects.

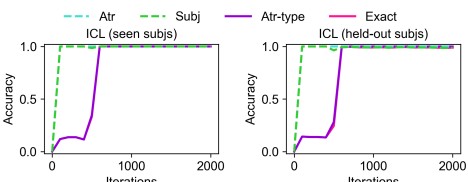

Figure 3: Finetuning the model pretrained on `PT` sequences, using `ICL` sequences with a subset of subjects, leads to good performance on `ICL` sequences with held-out query subjects.

> **Finding 2**: Finetuning on `ICL` sequences with a subset of subjects leads to generalization on `ICL` sequences with held-out subjects.

**Mechanism of Transfer.** Why does finetuning on a subset of subjects enable generalization to the rest? We posit that pretraining and finetuning play distinct but complementary roles. During pretraining, the model *acquires factual knowledge*: given a `PT` subsequence containing $s_i$, the model learns to decode the attribute type $\ell$ from the grammar statistics and predict the corresponding attribute $a_i^\ell$. During finetuning, the model does not learn new subject-attribute associations; rather, it learns a new *access mechanism:* inferring the attribute type implicitly from the attribute tokens in the context, rather than from explicit grammar cues. Because there is a shared structure between every $s_i$ and its type-$\ell$ attribute $a_i^\ell$ (across $i \in [M]$), the model can be finetuned on a subset of subjects to learn this implicit inference mechanism and generalize to held-out subjects.

**FT on `ICL+Grm` using a subset of subjects.** We next ask whether finetuning on `ICL` sequences is necessary to induce implicit attribute-type inference, or whether other distributions can achieve the same effect. To investigate this, we finetune the pre-trained model on `ICL+Grm` sequences with short, variable grammar length $S_{\mathrm{ft}}$. Specifically, we use a randomly sampled $S_{\mathrm{ft}} \in \{1, \ldots, 5\}$ to generate each `ICL+Grm` sequence. Due to the short grammar length, the sequence statistics are insufficient to reliably encode the attribute type, encouraging the model to instead infer it implicitly from the attribute tokens in the context. In this sense, these sequences serve the same purpose as `ICL` sequences.

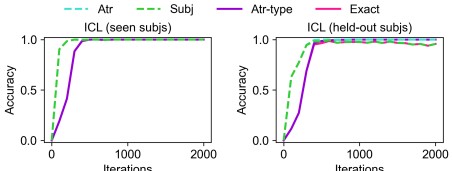

Figure 4: Finetuning the model pretrained on `PT` sequences, using `ICL+Grm` sequences with short, variable grammar length with a subset of subjects, leads to good performance on `ICL` sequences with held-out query subjects.

However, there remains a format distribution shift: during finetuning, attributes always follow a grammar token, whereas in evaluation `ICL` sequences, attributes directly follow subjects. The model must therefore learn to bridge this format gap at test time. Using variable grammar lengths prevents the model from overfitting to a fixed positional offset, encouraging it to rely on the context to infer the position of the next attribute token. Fig. 4 shows that the model successfully generalizes to `ICL` sequences for both seen and held-out subjects, as it is finetuned on `ICL+Grm` sequences.

> **Finding 3**: Finetuning on `ICL+Grm` sequences with short, variable grammar length using a subset of subjects leads to out-of-distribution generalization on `ICL` sequences with held-out subjects.

## 4 REPRESENTATIONAL ANALYSIS

We now analyze the model's internal representations to gain insight into the mechanism underlying contextual recall. Specifically, we investigate whether finetuning on `ICL+Grm` sequences induces the formation of a low-dimensional representation that encodes the shared attribute type $\ell$ from the in-context examples.

Let $X_{\ell,t} := [s_{i_0}, a_{i_0}^\ell, [\text{sep}], s_{i_1}, a_{i_1}^\ell, [\text{sep}], \ldots, s_{i_{t+1}}]$ denote an `ICL` sequence for an attribute type $\ell$, with $t \in [N]$ demonstrations. For each $\ell \in [L]$, we sample $K$ such sequences, denoted $X_\ell^k$ for $k \in [K]$. Let $f_j(\cdot)$ denote the model's representation at layer $j$ at the last token position. For fixed $j$ and $t$, we measure the cosine similarity for inter- and intra-task representations, averaged across the $K$ contexts for each pair $\ell, \ell' \in [L]$. Additionally, we quantify the representation clustering strength, $\bar{S}_j^t \in [-1, 1]$, in terms of a clustering metric using $1 - \cos(\cdot, \cdot)$ as the distance and attribute-type $\ell$ as the cluster label (see App. D for details). A high $\bar{S}^t$ indicates that representations of sequences with shared attribute type are tightly clustered and well-separated from those of different attribute types.

Fig. 5 shows the model's performance on `ICL` sequences with held-out subjects, alongside the representation clustering strength $\bar{S}^t$ computed from layer-2 attention representations, as the number of demonstrations $t$ increases. (Results for other layers are included in App. D.) We consider two settings: `Div`$\approx 0.2$ (top) and `Div`$\approx 0.5$ (bottom). In both cases, accuracy and clustering strength initially improve as $t$ is increased, and eventually saturate. This is also corroborated by the inset figures, which visualize the averaged cosine similarity for inter- and intra-task representations $\bar{C}^t(\ell, \ell')$ across attribute types $\ell, \ell' \in [L]$, after $t \in \{0, 2, 10\}$ demonstrations. This confirms that the finetuned model aggregates information from multiple examples to form a stable representation of the attribute type.

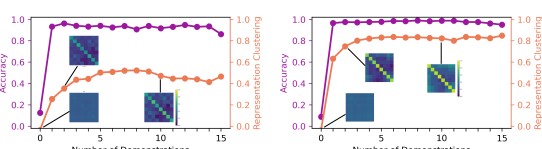

Figure 5: Comparison of accuracy on `ICL` sequences with held-out subjects, and layer-2 attention representation clustering strength (see text for details) for a finetuned model, as the number of demonstrations is increased, using `Div` $\approx 0.2$ (top) and `Div` $\approx 0.5$ (bottom). Inset figures visualize the averaged cosine similarity for inter- and intra-task representations. In both cases, both performance and clustering strength improve with number of demonstrations.

Interestingly, while accuracy is comparable in the two settings, higher `Div` during pretraining leads to stronger representation clustering. This suggests that the separation between the Markov chains (determined by `Div`) that encode attribute type information during pretraining, is reflected in the task vector separability after finetuning.

> **Finding 4**: The finetuned model encodes attribute type information from in-context examples in layer-2 attn. representations. Clustering strength increases with both the number of in-context examples and the separation between the attribute-specific Markov chains during pretraining.

## 5 CONCLUSION

We studied contextual recall, a form of ICL that requires models pretrained to acquire knowledge about various subjects and associated attributes, to recall a specific attribute for a query subject by implicitly inferring the relevant attribute type based on in-context examples. Our results give insights into the complementary roles of pretraining and finetuning in enabling in-context reasoning involving learned knowledge. Important directions for future work include characterizing the finetuning dynamics in the minimal setting we study in Appendix C, and investigating how incorporating new knowledge via further finetuning might impact the learned contextual recall capability.

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

# Contents

## A  Related Work

Prior work has leveraged controlled, synthetic settings to understand both ICL and factual recall in transformers trained from scratch, which we discuss below.

**Factual Recall.** Recent studies have utilized controlled synthetic setups to systematically explore how transformers acquire and recall factual associations. Allen-Zhu & Li (2023) and Zucchet et al. (2025) employ synthetic biography datasets, where each entry contains information about an individual or subject in the form of multi-sentence paragraphs. Each sentence associates some fact or attribute with an individual or subject, via a sentence template that contains information about the relation or attribute type, such as a person's birthplace, in the linguistic structure. They consider a fixed set of templates for each attribute type. Zucchet et al. (2025) evaluate factual knowledge (*i.e.*, correct attribute token prediction) using similar biography entries that are generated with templates from a held-out set, and identify a stage-wise learning dynamic where the model first learns to predict some attribute token, and then the correct attribute. We also observe similar stage-wise learning in our setup during pretraining in Section 2. In Allen-Zhu & Li (2023), the model is evaluated on question-answering (QA) formats, which constitutes a distribution shift from the biography-style format. Similar to our results in Section 3, they find that while pretraining is insufficient, finetuning (on QA) with a subset of subjects enables generalization on held-out subjects. Crucially, however, their QA prompts still contain explicit relation or attribute type information (e.g., "What is [subject]'s city of birth?"), whereas our contextual recall task constitutes a more substantial distribution shift as the context does not contain explicit relation cues. Investigating whether we see similar results as in

Section 3 for our synthetic setup with synthetic biography data used in these studies would be an interesting direction for future work.

A key differentiator across these frameworks is how "relation" information (the attribute type) is represented. Nichani et al. (2024) adopts an abstracted triplet-based approach where the relation is a single, dedicated token. In contrast, Behnia et al. (2025) omits the relation token entirely, using a Markov chain-based grammar setup to study how sequence statistics affect generalization; importantly, in their work, the facts are independent of the grammar. Our synthetic framework bridges these approaches: similar to Behnia et al. (2025), we model templates using Markov chains, but in contrast to their work, and similar to the other prior studies, we retain the relation information by associating specific Markov chains with specific attribute types.

For mechanistic analysis, we adopt an abstracted setup similar to Nichani et al. (2024) and compare our attention-only construction for factual recall (Proposition 1) with theirs. In both constructions, the relation heads perform a similar role: they attend to the (most recent) relation token in the context to output the sum of all attributes of the relevant type. However, the subject heads behave differently. While the construction in Nichani et al. (2024) uses subject heads to boost the attributes associated with the subject present in the context, our construction utilizes subject heads to suppress all attributes not associated with the subject. We also present experimental validation for our construction.

Notably, in prior works focusing on factual recall, the context the context contains explicit information (e.g., specific words, or linguistic patterns) required to recall specific learned facts, and does not require the model to perform any implicit inference (e.g., what fact is relevant based on the in-context examples) that is characteristic of ICL. In contrast, our work investigates contextual recall, a more challenging task compared to factual recall, that requires the model to infer the attribute type implicitly from given in-context examples.

**In-Context Learning.** Several recent studies have leveraged controlled synthetic environments to analyze how Transformers learn in-context when trained from scratch. A common approach involves training models on well-defined function classes, most notably linear regression (Garg et al., 2022; Raventos et al., 2023; Akyürek et al., 2023; von Oswald et al., 2022; Ahn et al., 2023; Zhang et al., 2025; Wu et al., 2023; Yang et al., 2025) and Markov chains (Park et al., 2025; Edelman et al., 2024; Rajaraman et al., 2024; Deora et al., 2025). These works often explore whether Transformers implement specific algorithms or functionalities—such as gradient descent (von Oswald et al., 2022; Ahn et al., 2023) or higher-order-algorithms (Fu et al., 2024) for linear regression, and induction heads for Markov chains (Edelman et al., 2024; Rajaraman et al., 2024; Chen et al., 2024b). Further research (Raventos et al., 2023; Park et al., 2025) has investigated task diversity, comparing *task retrieval* and *task learning* modes of ICL (Pan et al., 2023). Additionally, research has explored transient dynamics to understand how these two modes evolve over the course of training (Singh et al., 2023; Carroll et al., 2025; Singh et al., 2025). Finally, a growing body of work also explores the training dynamics of in-context learning by examining the optimization dynamics of linear regression (Zhang et al., 2025; 2023; 2024) in both one-layer linear attention and softmax attention models (Chen et al., 2024a).

A defining characteristic of the aforementioned studies is that the training and inference formats are identical; the model is evaluated on the same sequence structures it encountered during training. In contrast, our work on contextual recall introduces a significant prompt-distribution shift. While pretraining focuses on instilling factual knowledge in the model via explicit grammar-based cues, the ICL evaluation requires the model to transition a novel format necessitating inference of the relevant attribute type implicitly from the in-context demonstrations. Therefore, unlike standard ICL setups that focus on function induction, our framework requires the model to bridge the gap between structured knowledge acquisition and implicit in-context reasoning.

In contrast to our work, Xie et al. (2022) investigate a distribution shift that is primarily compositional rather than structural. In their framework, the shift occurs when a model pretrained on long, continuous documents is prompted with a sequence of independent, i.i.d. examples. While they frame ICL as a statistical process of implicit Bayesian inference, our work provides a mechanistic perspective discussed in Section 4 and Appendix C.

# B  DATA GENERATION PROCESS

To study contextual recall in transformers, we define three different types of sequences for pretraining, finetuning, and/or evaluation. The first, `PT`, is designed to instill factual knowledge, while the other two, `ICL` and `ICL+Grm`, are used to evaluate or facilitate how the model leverages such knowledge for in-context reasoning tasks.

We first introduce some useful notation. Let $\mathcal{S} = \{s_j\}_{j=1}^{M}$ denote the set of $M$ subjects (e.g., landmarks such as `Parthenon` ($j = 1$), `Colosseum` ($j = 2$)). Let the set of unique attributes be denoted as $\mathcal{U} = \cup_{\ell=1}^{L} \mathcal{U}_\ell$, where $\mathcal{U}_\ell = \{u_i^\ell\}_{i=1}^{M_\ell}$, and $\ell$ indexes the attribute type (e.g., $\ell = 1$ for "country", $\ell = 2$ for "year built"), and $M_\ell$ denotes the number of unique values for index $\ell$ (e.g., $\{$`Greece`, `Italy`, `Canada`, ...$\}$ for "country"). Next, let $\mathcal{A} = \cup_{j=1}^{M} \{a_j^1, \ldots, a_j^L\}$ denote the set of attributes assigned to the subjects, where $a_j^\ell$ is the type-$\ell$ attribute of subject $j$ (e.g., $a_2^1 =$ `Italy`). Each subject $s_j$ has one attribute per type, giving $L$ attributes per subject. Let $\mathcal{G} = \{\tilde{g}_1, \ldots, \tilde{g}_G\}$ denote a set of $G$ grammar tokens, and [sep] denote the separator token. The full vocabulary is $\mathcal{V} = \mathcal{S} \cup \mathcal{U} \cup \mathcal{G} \cup \{[\text{sep}]\}$, with size $V = M + \sum_{\ell=1}^{L} M_\ell + G + 1$. Let $\text{Unif}(\cdot)$ denote the uniform distribution, and $||$ denote concatenation.

With this notation established, we now describe the data generation process. The key building block is a *subject-grammar-attribute subsequence*. For subject $s_j$, attribute type $\ell$, and grammar sequence length $S$, this subsequence is denoted as $X_j^\ell = [s_j, g_1, \ldots, g_S, a_j^\ell]$ (e.g., [`Parthenon`, *one*, *of*, *the*, *wonders*, ..., *is in*, `Greece`]). The grammar sequence $g_{1:S}$ is generated using a first-order Markov chain specific to attribute type $\ell$. That is, $p(g_t = \tilde{g}|g_{t-1}, \ldots, g_1) = p(g_t = \tilde{g}|g_{t-1})$ for all $\tilde{g} \in \mathcal{G}$, with the first token drawn uniformly at random. For each attribute type $\ell \in [L]$, we sample a row-stochastic transition matrix $\text{TM}_\ell \in \mathbb{R}^{G \times G}$, where each row is drawn independently from a Dirichlet prior with parameter $\boldsymbol{\alpha}$. Note that since each attribute type has its own Markov chain, the bigram statistics of the grammar sequence $g_{1:S}$ implicitly encode information about the attribute type $\ell$.

We now define the three types of sequences used for pretraining, finetuning, and/or evaluation; see Fig. 1 for an illustration.

**`PT` Sequences (Pretraining).**   To instill factual knowledge in the model, we use `PT` sequences which contain information about a specific subject and its associated attributes (analogous to a short encyclopedia entry for a given landmark). Let $N_{\text{tr}}$ denote the number of subsequences (subject-grammar-attribute tuples) in each sequence. To generate a `PT` sequence, we first sample a subject $j \sim \text{Unif}([M])$, then draw $N_{\text{tr}}$ attribute types $\ell_{1:N_{\text{tr}}} \sim \text{Unif}([L])^{N_{\text{tr}}}$, and sample each grammar subsequence $g_{1:S}^i$ from the corresponding Markov chain $\text{TM}_{\ell_i}$. The resulting sequence is $\tilde{X} = X_j^{\ell_1}||[\text{sep}]||X_j^{\ell_2} \ldots X_j^{\ell_{N_{\text{tr}}}}$, with sequence length $T = (S + 3)N_{\text{tr}} - 1$. To model generic text that does not convey factual information, we also include grammar-only subsequences: with probability $p_G$, each subject-grammar-attribute subsequence is replaced by a grammar-only subsequence $g_{1:S+2}$, generated from a separate fixed Markov chain (with transition matrix sampled from a Dirichlet prior). This gives the final sequence $X$.

**`ICL` Sequences (Finetuning and Evaluation).**   To test for contextual recall, we use `ICL` sequences that contain subject-attribute pairs with a shared attribute type across different subjects—mirroring the evaluation format from the introduction (e.g., [`Niagara Falls`, `Canada`, [sep], `Colosseum`, `Italy`, [sep], `Parthenon`, ]). Let $N$ denote the number of in-context demonstrations. First, we sample an attribute type $\ell \sim \text{Unif}([L])$, and $N$ subjects $j_{1:N+1} \sim \text{Unif}([M])^{N+1}$. The resulting `ICL` sequence is $X = [s_{j_1}, a_{j_1}^\ell, [\text{sep}], \ldots, s_{j_{N+1}}, a_{j_{N+1}}^\ell]$. At evaluation, the model observes $N$ subject-attribute pairs and must predict the attribute $a_{j_{N+1}}^\ell$ for the final subject $s_{j_{N+1}}$, without any grammar cues indicating the attribute type.

**`ICL+Grm` Sequences (Finetuning).**   We also use `ICL+Grm` sequences for finetuning. As the name suggests, these sequences are similar to `ICL` sequences (subject-attribute pairs sharing a common attribute type across different subjects), but they also contain grammar tokens between the subject and attribute. (As we saw in Section 3, these sequences help instill the implicit inference ability required to succeed at contextual recall on `ICL` sequences.) Let $S_{\text{ft}}$ denote the grammar sequence length used in each subsequence, and $N_{\text{ft}}$ denote the number of subsequences. We sample attribute type $\ell \sim \text{Unif}([L])$ and subjects $j_{1:N_{\text{ft}}+1} \sim \text{Unif}([M])^{N_{\text{ft}}+1}$, then generate grammar

subsequences $g^i_{1:S_{\mathrm{ft}}}$ from the corresponding Markov chain $\mathrm{TM}_\ell$. The resulting `ICL+Grm` sequence is $X = X^\ell_{j_1}||[\mathrm{sep}]||X^\ell_{j_2}\ldots X^\ell_{j_{N_{\mathrm{ft}}}}$. Note that `ICL` sequences are a special case of `ICL+Grm` with $S_{\mathrm{ft}} = 0$.

## C  Mechanistic Analysis in a Simpler Setting

To gain mechanistic insight into how pretraining and finetuning contribute to contextual recall, we study an analytically tractable setting that preserves the qualitative behavior observed in Findings 1-2 of Section 2. Specifically, we simplify in two ways: i) we encode attribute type information in explicit "relation" tokens rather than grammar sequence statistics, and ii) we use a one-layer attention-only model. Below, we first describe our setup, then present constructions for accurately predicting the final attribute token on both the `PT` and `ICL` sequences, and then present experimental validation for the constructions.

**Data Setting.** Let $\mathcal{R} := \{r_1, \ldots, r_L\}$ denote a set of relation tokens, one per attribute type. The `PT` sequences are generated in a similar manner to Appendix B, with a modification to the grammar subsequences. For each subsequence $g_{1:S}$, we first sample a position $i \sim \mathrm{Unif}([S])$ and a relation token $r \sim \mathrm{Unif}(\mathcal{R})$, and set $g_i = r$. Then, we sample the remaining entries independently as $g_j \sim \mathrm{Unif}(\mathcal{G})$ for all $j \neq i$. This setup is similar to the synthetic setup used to study factual recall in Nichani et al. (2024). The `ICL` sequences are generated as in Appendix B.

**Model.** We consider a single-layer attention-only model with fixed relative positional encoding added to the key inputs. The output at position $t$ is defined as follows:

$$g^t_h(\boldsymbol{X}) = \boldsymbol{X}^\top_{1:t}\boldsymbol{\varphi}((\boldsymbol{X}_{1:t} + \boldsymbol{P}_{\mathrm{T}-t+1:\mathrm{T}})^\top \boldsymbol{W}^h_{KQ}\boldsymbol{x}_t)$$

$$f^t_\Theta(\boldsymbol{X}) = \sum_{h=1}^H \underbrace{\boldsymbol{W}^h_{OV} g^t_h(\boldsymbol{X})}_{f^t_h(\boldsymbol{X})}. \tag{1}$$

where $\boldsymbol{X} \in \mathbb{R}^{\mathrm{T}\times d}$ denotes the input sequence, $\boldsymbol{x}_t \in \mathbb{R}^d$ denotes the $t^{\mathrm{th}}$ token, $\boldsymbol{P} = [\boldsymbol{h}(p_{-\mathrm{T}+1}), \ldots, \boldsymbol{h}(p_0)]$ denotes the positional encodings , $\boldsymbol{\varphi}(\cdot)$ denotes the softmax, and for convenience, we define $\boldsymbol{W}^h_{OV} = (\boldsymbol{W}^h_O)^\top \boldsymbol{W}^h_V$, $\boldsymbol{W}^h_{KQ} = (\boldsymbol{W}^h_K)^\top \boldsymbol{W}^h_Q$, for $h \in [H]$ using output,value, key, query weight matrices $\boldsymbol{W}^h_O, \boldsymbol{W}^h_V, \boldsymbol{W}^h_K, \boldsymbol{W}^h_Q \in \mathbb{R}^{d_h \times d}$. We use the shorthand $f(\cdot) = f^\mathrm{T}_\Theta(\cdot)$ and $g_h(\cdot) = g^\mathrm{T}_h(\cdot)$ to denote the outputs at the last token position. The model prediction is

$$v^* = \arg\max_{v\in\mathcal{V}} \boldsymbol{h}(v)^\top f(\boldsymbol{X}),$$

where $\boldsymbol{h}(v) \in \mathbb{R}^d$ denotes the embedding for token $v$. We use fixed embeddings and tied unembeddings. We consider one-hot embeddings and positional encodings and set $d = V + \mathrm{T}$, so that these subspaces are orthogonal.

**Construction for `PT` Sequences.** We first investigate whether an attention-only model is expressive enough to perform factual recall on `PT` sequences of the form

$$\boldsymbol{X} = \boldsymbol{X}^{\ell_1}_{\bar{j}}||[\boldsymbol{h}(a^{\ell_1}_{\bar{j}})]||\boldsymbol{X}^{\ell_2}_{\bar{j}}\ldots \boldsymbol{X}^{\ell_{N_{\mathrm{tr}}}}_{\bar{j}}, \text{ where}$$

$$\boldsymbol{X}^{\ell_1}_{\bar{j}} = [\boldsymbol{h}(s_{\bar{j}}), \boldsymbol{h}(g_1), ..., \boldsymbol{h}(r_{\ell_1}), ..., \boldsymbol{h}(g_S), \boldsymbol{h}([\mathrm{sep}])]. \tag{2}$$

Here, $\boldsymbol{x}_\mathrm{T} = \boldsymbol{h}([\mathrm{sep}])$, and the correct last-token prediction is $\boldsymbol{h}(a^{\ell_{N_{\mathrm{tr}}}}_{\bar{j}})$. We first show that there exists an attention-only model capable of perfectly predicting the next token for such sequences. For simplicity, we focus on last-token prediction here; the construction extends to predictions at any position (see App. E.1).

**Proposition 1** (Informal)**.** *Consider the input $\boldsymbol{X}$ in Eq. (2). There exists a single-layer attention-only model such that when $\|\boldsymbol{W}^h_{KQ}\| \to \infty$ , correctly predicts the last token—returning the attribute $a^{\ell_{N_{\mathrm{tr}}}}_{\bar{j}}$ corresponding to the sequence's subject $s_{\bar{j}}$ and attribute type $\ell_{N_{\mathrm{tr}}}$.*

*Proof Sketch.* We present a construction with a 3-head model. At a high level, two heads, which we call the **relation** and **subject** heads, are responsible for the prediction at the target position (following $\boldsymbol{h}([\mathrm{sep}])$), and the third **grammar** head for other cases. For simplicity, we present the construction

here assuming that $f_{\text{grm}}(\boldsymbol{X}) = 0$, since the outputs from this head do not affect the conclusions in this case, as shown in the full proof in Appendix E.1.

First, the **relation head** attends to the most recent relation token $r_{\ell_{N_{\text{tr}}}}$ and maps it to the sum of *all* attributes of type $\ell_{N_{\text{tr}}}$. Specifically,

$$\boldsymbol{W}_{KQ}^{\text{rel}} = \beta\Big(\sum_\ell \boldsymbol{h}(r_\ell) + \boldsymbol{p}\Big)\boldsymbol{h}([\text{sep}])^\top, \quad \boldsymbol{W}_{OV}^{\text{rel}} = \sum_\ell \sum_j \boldsymbol{h}(u_j^\ell)\boldsymbol{h}(r_\ell)^\top,$$

where $\boldsymbol{p} := \sum_{i=1}^{S+2} \boldsymbol{h}(p_{-i})$ With $\beta \to \infty$, the head's outputs become $g_{\text{rel}}(\boldsymbol{X}) = \boldsymbol{h}(r_{\ell_{N_{\text{tr}}}})$, i.e., the most recent relation token in the sequence, and $f_{\text{rel}}(\boldsymbol{X}) = \sum_j \boldsymbol{h}(u_j^{\ell_{N_{\text{tr}}}})$, *i.e.*, all attributes of type $\ell_{N_{\text{tr}}}$.

On the other hand, the **subject head** attends to the subject token $s_{\bar{j}}$ and filters out irrelevant attributes (those not associated with $s_{\bar{j}}$). Specifically,

$$\boldsymbol{W}_{KQ}^{\text{subj}} = \beta\Big(\sum_j \boldsymbol{h}(s_j)\Big)\boldsymbol{h}([\text{sep}])^\top, \quad \boldsymbol{W}_{OV}^{\text{subj}} = -\sum_j \Big(\sum_{j' \neq j, \ell} \boldsymbol{h}(a_{j'}^\ell)\Big)\boldsymbol{h}(s_j)^\top.$$

Then, with $\beta \to \infty$, this head outputs $g_{\text{subj}}(\boldsymbol{X}) = \boldsymbol{h}(s_{\bar{j}})$, the subject token in the sequence, and $f_{\text{subj}}(\boldsymbol{X}) = -\Big(\sum_u \boldsymbol{h}(u) - \sum_\ell \boldsymbol{h}(a_{\bar{j}}^\ell)\Big)$, *i.e.*, the negative of all attributes that are not associated with subject $s_{\bar{j}}$.

Combining these outputs, the model isolates the specific attribute $a_{\bar{j}}^{\ell_{N_{\text{tr}}}}$, i.e., the attribute reinforced by both heads: $f(\boldsymbol{X}) = \sum_j \boldsymbol{h}(u_j^{\ell_{N_{\text{tr}}}}) + \sum_\ell \boldsymbol{h}(a_{\bar{j}}^\ell) - \sum_u \boldsymbol{h}(u)$, yielding the correct prediction $v^* = a_{\bar{j}}^{\ell_{N_{\text{tr}}}}$.

**Construction for `ICL` Sequences.** Next, consider `ICL` sequences of the form

$$\boldsymbol{X} = [\boldsymbol{h}(s_{j_1}), \boldsymbol{h}([\text{sep}]), \boldsymbol{h}(a_{j_1}^{\bar{\ell}}), ..., \boldsymbol{h}(s_{j_{N_{\text{ft}}+1}}), \boldsymbol{h}([\text{sep}])], \tag{3}$$

with correct last token prediction $\boldsymbol{h}(a_{j_{N_{\text{ft}}+1}}^{\bar{\ell}})$. We show that there exists an attention-only model that perfectly predicts the last token for such sequences.

**Proposition 2** (Informal). *Consider the input $\boldsymbol{X}$ in Eq.* (3). *There exists single-layer attention-only model such that when $\|\boldsymbol{W}_{KQ}^h\| \to \infty$, it correctly predicts the last token, returning the attribute $a_{j_{N_{\text{ft}}+1}}^{\bar{\ell}}$ corresponding to the query subject $s_{j_{N_{\text{ft}}+1}}$ and the shared attribute type $\bar{\ell}$.*

*Proof Sketch.* We adapt the construction from Proposition 1 with minimal changes, but with one crucial modification that reflects the role of finetuning. The **subject** head operates similarly, attending to the query subject and filtering out attributes not associated with it. The key difference lies in the **relation** head: since `ICL` sequences contain no explicit relation tokens, this head must now *infer the attribute type implicitly* by attending to the attribute tokens in the context and mapping them to all attributes of the same type $\bar{\ell}$.

Specifically, for the **subject** head, we set

$$\boldsymbol{W}_{KQ}^{\text{subj}} = \beta\Big(\sum_j \boldsymbol{h}(s_j) + \boldsymbol{h}(p_{-1})\Big)\boldsymbol{h}([\text{sep}])^\top, \quad \boldsymbol{W}_{OV}^{\text{subj}} = -\sum_j \Big(\sum_{j' \neq j, \ell} \boldsymbol{h}(a_{j'}^\ell)\Big)\boldsymbol{h}(s_j)^\top.$$

Note that as compared to the construction for `PT` sequences (Proposition 1), $\boldsymbol{W}_{OV}^{\text{subj}}$, which contains subject-attribute information, is unchanged. The only difference here is that $\boldsymbol{W}_{KQ}^{\text{subj}}$ now contains $\boldsymbol{h}(p_{-1})$, so that when $\beta \to \infty$, $g_{\text{subj}}(\boldsymbol{X}) = \boldsymbol{h}(s_{j_{N_{\text{ft}}+1}})$, *i.e.*, the query subject, and hence, $f_{\text{subj}}(\boldsymbol{X}) = -\Big(\sum_u \boldsymbol{h}(u) - \sum_\ell \boldsymbol{h}(a_{j_{N_{\text{ft}}+1}}^\ell)\Big)$.

We now discuss the **relation** head. Since in our experiments, we finetune with a subset of subjects, let $\mathcal{S}' \subset \mathcal{S}$ denote a subset of subjects. Next, for each attribute type $\ell$, let $\mathcal{U}_\ell' := \cup_{j \in \mathcal{S}'}\{a_j^\ell\}$ denote the set of unique attributes seen during finetuning. With this notation established, we set

$$\boldsymbol{W}_{KQ}^{\text{rel}} = \beta \sum_\ell \sum_{u \in \mathcal{U}_\ell'} \boldsymbol{h}(u)\boldsymbol{h}([\text{sep}])^\top, \quad \boldsymbol{W}_{OV}^{\text{rel}} = \sum_\ell \Big(\sum_j \boldsymbol{h}(u_j^\ell)\Big)\Big(\sum_{u \in \mathcal{U}_\ell'} \boldsymbol{h}(u) + \boldsymbol{h}(r_\ell)\Big)^\top.$$

In contrast to the construction for `PT` sequences, where both $\boldsymbol{W}_{KQ}^{\mathrm{rel}}$ and $\boldsymbol{W}_{OV}^{\mathrm{rel}}$ rely on the relation tokens $\boldsymbol{h}(r_\ell)$, in this case, they rely on the attributes seen during finetuning, *i.e.*, $\sum_{u \in \mathcal{U}'_\ell} \boldsymbol{h}(u)$. In this case, when $\beta \to \infty$, $g_{\mathrm{rel}}(\boldsymbol{X}) = \frac{1}{N_{\mathrm{ft}}} \sum_i \boldsymbol{h}(a_{j_i}^{\bar{\ell}})$, *i.e.*, the average of the attributes that appear in the context, and $f_{\mathrm{rel}}(\boldsymbol{X}) = \sum_j \boldsymbol{h}(u_j^{\bar{\ell}})$, retrieving the sum of all attributes of the shared type $\bar{\ell}$.

Finally, by combining the outputs from the two heads, the model isolates the specific attribute $a_{j_{N_{\mathrm{ft}}+1}}^{\bar{\ell}}$ that is encouraged by both, *i.e.* $f(\boldsymbol{X}) = \sum_j \boldsymbol{h}(u_j^{\bar{\ell}}) + \sum_\ell \boldsymbol{h}(a_{j_{N_{\mathrm{ft}}+1}}^{\ell}) - \sum_u \boldsymbol{h}(u)$. Then, the final prediction $v^* = a_{j_{N_{\mathrm{ft}}+1}}^{\bar{\ell}}$.

Note that, for the **relation** head to output all attributes of the same type as the *attributes* that appear in the context, for any type $\ell$, the set of attributes seen during finetuning ($\mathcal{U}'_\ell$) can be much smaller than the full set $\mathcal{U}_\ell$. This helps explain why finetuning on a subset of subjects can enable generalization on held-out subjects.

**Experimental Validation.** We present experimental evidence to corroborate our theoretical constructions, using a 1-layer 3-head attention-only model (see App. D for details).

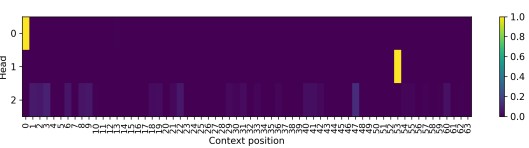

Figure 6: Attention scores for each head across a `PT` sequence at the end of pretraining. Head 0 attends to the first subject, while head 1 attends to the most recent relation token, as predicted by Proposition 1.

Figure 7: Cosine similarity between the actual outputs of each head and the outputs $f_{\mathrm{subj}}(\boldsymbol{X})$ (left) and $f_{\mathrm{rel}}(\boldsymbol{X})$ (right) from our construction in Proposition 1. Head 0 output closely matches $f_{\mathrm{subj}}(\boldsymbol{X})$, while head 1 matches $f_{\mathrm{rel}}(\boldsymbol{X})$.

*Validation of Pretraining Mechanism.* We first train the model using `PT` sequences (Eq. (2)) with the next-token prediction objective. In Fig. 6, we visualize the attention scores for each head across a `PT` sequence, and find that the heads specialize into distinct roles consistent with Proposition 1. We find that head 0 attention score concentrates on the first subject token, and it outputs $\boldsymbol{h}(s_{\bar{j}})$, matching the role of $g_{\mathrm{subj}}(\boldsymbol{X})$ in our construction, while head 1 attends to the most recent relation token, *i.e.*, it outputs $\boldsymbol{h}(r_{\ell_{N_{\mathrm{tr}}}})$, consistent with $g_{\mathrm{rel}}(\boldsymbol{X})$. Further, in Fig. 7, we compute the cosine similarity between the head outputs and theoretical outputs of the subject and relation heads specified by our construction. Specifically, we report the cosine similarity with $f_{\mathrm{subj}}(\boldsymbol{X})$, *i.e.*, negative sum of all attributes not associated with the subject $s_{\bar{j}}$ (left subplot) and $f_{\mathrm{rel}}(\boldsymbol{X})$, *i.e.*, sum of all attributes of type $\ell_{N_{\mathrm{tr}}}$ (right), averaged across several sequences. We find that head 0 output closely matches $f_{\mathrm{subj}}(\boldsymbol{X})$, while head 1 matches $f_{\mathrm{rel}}(\boldsymbol{X})$. We designate these heads as subject and relation heads, respectively. Together, these results present experimental validation of our construction for `PT` sequences.

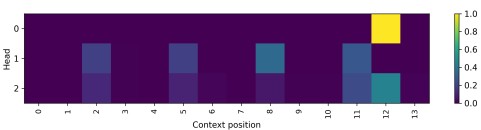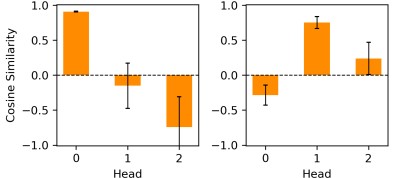

Figure 8: Attention scores for each head across an `ICL` sequence at the end of finetuning. Head 0 attends to the most recent subject, while head 1 attends to the attribute tokens in the sequence, as predicted by Proposition 2.

Figure 9: Cosine similarity between the actual outputs of each head and the outputs $f_{\mathrm{subj}}(\boldsymbol{X})$ (left) and $f_{\mathrm{rel}}(\boldsymbol{X})$ (right) from our construction in Proposition 2. Head 0 output closely matches $f_{\mathrm{subj}}(\boldsymbol{X})$, while head 1 matches $f_{\mathrm{rel}}(\boldsymbol{X})$.

*Validation of ICL Mechanism.* We finetune the model on `ICL` sequences (Eq. (3)) with the last-token prediction objective, using $50\%$ of the total subjects (Fig. 16 in the App. shows that the model generalizes on held-out subjects). Visualizing the attention scores on an `ICL` sequence in Fig. 8

reveals that the model repurposes its heads for the new task, consistent with Prop. 2. We find that the head 0 (subject head) attends to the most recent/query subject token, while head 1 (relation head) attends to the attribute tokens in the context, *i.e.*, it outputs a combination of attributes of type $\bar{\ell}$. Further, Fig. 9 confirms that the outputs of these heads closely match the theoretical outputs specified by our construction for `ICL` sequences: head 0 (subject head) outputs negative sum of all attributes not associated with the subject $s_{j_{N_{\mathrm{ft}}}}$ (left subplot), while head 1 (relation head) outputs the sum of all attributes of type $\bar{\ell}$ (right).

# D  ADDITIONAL RESULTS AND DETAILS OF EXPERIMENTAL SETTINGS

## D.1  EFFECT OF DATA DISTRIBUTION

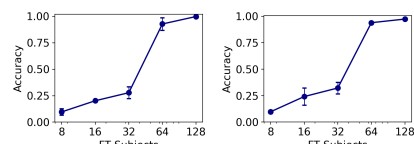

Here, we first investigate whether the format distribution shift inherent in finetuning with `ICL+Grm` sequences incurs a cost in terms of sample efficiency compared to direct finetuning with `ICL` sequences. In Fig. 10, we compare the `ICL` performance on held-out subjects as we vary the number of finetuning subjects. Interestingly, we find that the sample complexity is comparable across both settings, with performance improving monotonically as the number of finetuning subjects increases.

Figure 10: Increasing the number of finetuning subjects with `ICL` (left) or `ICL+Grm` (right) sequences improves the model's performance on `ICL` sequences with held-out query subjects.

Having established (in Section 3) that finetuning on `ICL+Grm` enables contextual recall, we now investigate how the properties of the pretraining data influence this capability. Unless stated otherwise, we use the same parameters as in Sections 2 and 3 and use `ICL+Grm` sequences with short, variable grammar length for finetuning.

Keeping other parameters fixed, we first consider the effect of varying the sequence length $S$ and the diversity between the Markov chains for different attribute types, quantified as $\mathrm{Div} := \min_{\ell,\ell'} \|\mathrm{TM}_\ell - \mathrm{TM}'_{\ell'}\|_1$, where $\|\cdot\|_1$ denotes the $\ell_1$-norm. Intuitively, a higher $\mathrm{Div}$ for a fixed sequence length implies that the grammar statistics for different attribute types are more distinct, making the attribute type easier to distinguish during pretraining. As shown in Fig. 11, increasing the sequence length $S$ and/or the Markov chain diversity $\mathrm{Div}$ used while pretraining, improves the model's performance both on the `PT` sequences, as well as on `ICL` sequences with held-out subjects after finetuning on `ICL+Grm` sequences with short, variable grammar length.

Next, in Fig. 12, we probe the effect of increasing the number of subjects $M$ used for pretraining. Increasing $M$ improves the performance on `ICL` sequences with held-out subjects after fine-tuning. Here, we use grammar sequence length $S = 80$, and consistent with Fig. 11, observe that using a sufficiently large $S$ for pretraining leads to comparable the final `ICL` performance across different levels of $\mathrm{Div}$. We summarize these results in our next finding.

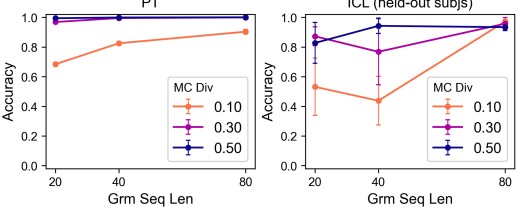

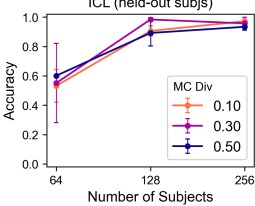

Figure 11: Increasing the sequence length $S$ and/or Markov chain diversity $\mathrm{Div}$ used while pretraining improves the model's performance on PT sequences as well as on `ICL` sequences with held-out subjects after finetuning on `ICL+Grm` sequences with short, variable grammar length.

Figure 12: Increasing the number of subjects $M$ used while pretraining improves the model's performance on `ICL` sequences with held-out subjects after finetuning on `ICL+Grm` sequences with short, variable grammar length.

> **Finding 5**: Increasing the number of subjects, the grammar sequence length, or the separation between the Markov chains for different attribute types used while pretraining improves the final performance on `ICL` sequences.

## D.2 DETAILS OF EXPERIMENTAL SETTINGS

We use a 2-layer 1-head GPT-2 type decoder-only transformer model (Karpathy) with embedding dimension 256. We train the model with AdamW optimizer (Loshchilov & Hutter, 2019) with learning rate $10^{-4}$, weight decay 0.001, and batch size 64, for both pretraining and finetuning.

For the experiments in Section 2, we set $M = 256, L = 8, N_{\text{tr}} = 5, M_1 = 256, M_2 = \cdots = M_8 = 32, S = 80$. We set the grammar-only subsequence probability $p_G = 0.2$, and separation between Markov chains `Div` $\approx 0.5$. To control for `Div`, we first randomly generate a large pool of transition matrices, and then use greedy selection to curate a subset of transition matrices that are assigned to each attribute type. We pretrain the model for $20k$ iterations.

For the experiments in Section 3, we use the same pretraining setting as in Section 2, and set $N = N_{\text{ft}} = 16$. We use 128 subjects for finetuning, unless stated otherwise. For experiments with `ICL+Grm` sequences, $S_{\text{ft}} \sim \text{Unif}(\{1, \ldots, 5\})$. In all cases, we finetune for $2k$ iterations. In Fig. 10, we compare the best performance across finetuning iterations.

For Figs. 10 to 12, the results are reported after averaging across two random initialization seeds.

For the experiments in Appendix D.1, the details are as follows. All results are reported at end of pretraining/finetuning. We consider `Div` $\approx 0.1, 0.3, 0.5$. In Fig. 11, we use fixed $M = 256$ and for $S = 20$, we use $S_{\text{ft}} \sim \text{Unif}(\{1, ..., 4\})$, while for $S = 40$ or $S = 80$, we use $S_{\text{ft}} \sim \text{Unif}(\{1, ..., 5\})$. In Fig. 12, we use fixed $S = 80, S_{\text{ft}} \sim \text{Unif}(\{1, ..., 5\})$.

The results in Fig. 5 (bottom) are reported with the same setting as Fig. 4. For the top plot, we only change `Div` $\approx 0.2$. We use 50 sequences for each attribute type.

The settings for the experiments in Appendix C are as follows. We train an attention-only model with $d_h = 256$ using AdamW optimizer with learning rate 0.001, weight decay 0.001, and batch size 64, for both pretraining ($20k$ iterations) and finetuning ($2k$ iterations). We use a cosine learning rate scheduler for pretraining. We set $M = 256, L = 8, N_{\text{tr}} = 5, M_1 = 256, M_2 = \cdots = M_8 = 32, S = 10, N = N_{\text{ft}} = 5$.

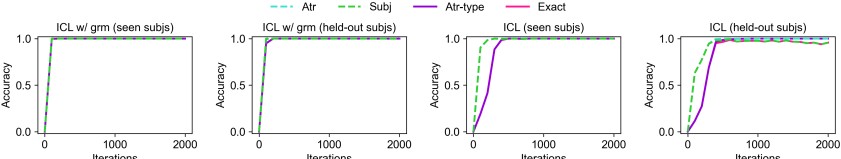

Figure 13: Performance of the model when finetuning with `ICL+Grm` sequences with a subset of subjects (same setting as Fig. 4), on `ICL+Grm` sequences with $S_{\text{ft}} = 1$ (left) and `ICL` sequences (right) with seen or held-out subjects. Finetuning with `ICL+Grm` sequences enables out-of-distribution generalization on `ICL` sequences with held-out subjects. Performance improves first on `ICL+Grm` and later on `ICL` sequences.

## D.3 DETAILS OF REPRESENTATIONAL ANALYSIS

Consider `ICL` sequences, which are of the form

$$X_\ell = [s_{i_0}, a^\ell_{i_0}, [\text{sep}], s_{i_1}, a^\ell_{i_1}, [\text{sep}], \ldots, s_{i_{N+1}}],$$

for a fixed attribute type $\ell$. We sample $K$ such sequences for each $\ell \in [L]$, denoted by $X^k_\ell$. Also, define $X_{\ell,t} := [s_{i_0}, a^\ell_{i_0}, [\text{sep}], s_{i_1}, a^\ell_{i_1}, [\text{sep}], \ldots, s_{i_{t+1}}]$, where $t \in [N]$ denotes the number of demonstrations.

Let $f_j(\cdot)$ denote the model's representation at layer-$j$. Consider fixed $j, t$, and for $\ell, \ell' \in [L]$, define

$$\bar{C}^t_j(\ell, \ell') = \frac{1}{K^2} \sum_{k,k'} \cos\big(f_j(X^k_{\ell,t}), f_j(X^{k'}_{\ell',t})\big).$$

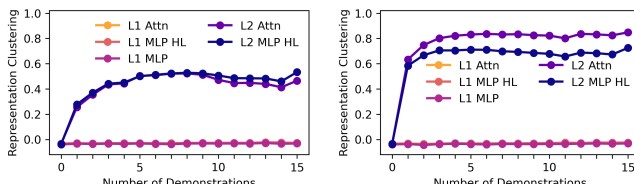

Figure 14: Comparison of clustering strength (see Section 4 for details) using representations from different layers of the finetuned 2-layer 1-head transformer with `ICL` sequences as inputs, as the number of demonstrations is increased (same setting as Fig. 5) with `Div` $\approx 0.2$ (left) and `Div` $\approx 0.5$ (right). We find that layer-2 attention representations cluster most strongly, while layer-1 representations exhibit no clustering based on the attribute type information in the context. Further, using higher `Div` while pretraining leads to stronger representation clustering after finetuning.

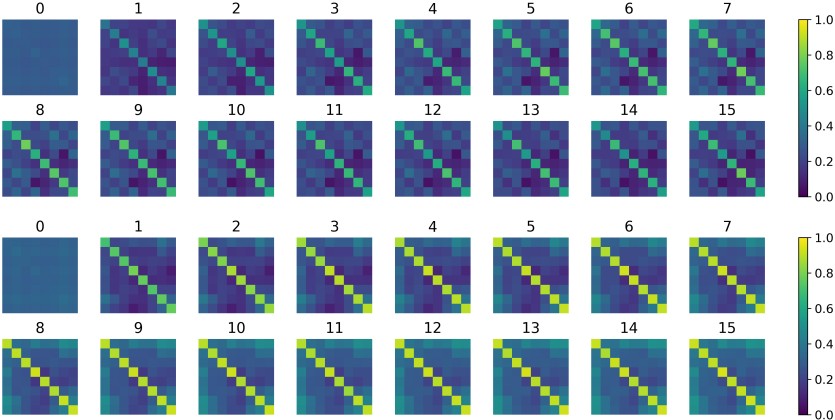

Figure 15: Each subfigure visualizes the cosine similarity for inter- and intra-task representations (from layer-2 attention layer of the finetuned model), $\bar{C}^t(\ell, \ell')$ (see Section 4 for details) across attribute types $\ell, \ell' \in [L]$, averaged over 50 sequences for each attribute type (same setting as Fig. 5; top: `Div` $\approx 0.2$, bottom: `Div` $\approx 0.5$). We find that as the number of demonstrations is increased (from 0 to 15), the representations of `ICL` sequences with the same attribute type get clustered together, and the clustering is stronger for higher `Div` (*i.e.*, more separated attribute-specific Markov shains).

If the model perfectly disentangles attribute types in its representations, then, for some layer $j$ and number of demonstrations $t$, we would have $\bar{C}_j^t(\ell, \ell') = \mathbb{1}\left[\ell = \ell'\right]$. Next, the clustering strength is quantified as follows. For fixed $j, t$, we subsume these and define $v_\ell^k := f_j^t(X_\ell^k)$. Each representation is assigned cluster label based on the attribute type $\ell$. Define the cosine distance $d(v, w) := 1 - \cos(v, w)$. For each point $v_\ell^k$, define the intra-cluster dissimilarity

$$a(k, \ell) = \frac{1}{K - 1} \sum_{k' \neq k} d(v_\ell^k, v_\ell^{k'}),$$

and the nearest other-cluster dissimilarity

$$b(k, \ell) = \min_{\ell' \neq \ell} \left( \frac{1}{K} \sum_{k'} d(v_\ell^k, v_{\ell'}^{k'}) \right).$$

The silhouette value for a sample is

$$s(k, \ell) = \frac{b(k, \ell) - a(k, \ell)}{\max\{a(k, \ell),\, b(k, \ell)\}}.$$

Finally, the silhouette score for layer $j$ after $t$ demonstrations are seen is

$$\bar{S}_j^t = \frac{1}{KL} \sum_\ell \sum_k s(k, \ell).$$

# E  ADDITIONAL DETAILS FOR APPENDIX C

## E.1  CONSTRUCTIONS FOR PT AND ICL SEQUENCES

Consider PT sequences of the form

$$\boldsymbol{X} = [\boldsymbol{h}(s_{\bar{i}}), \boldsymbol{h}(g_1), ..., \boldsymbol{h}(r_{\ell_1}), ..., \boldsymbol{h}([\text{sep}]), \boldsymbol{h}(a_{\bar{i}}^{\ell_1}), \boldsymbol{h}(s_{\bar{i}}), ..., \boldsymbol{h}([\text{sep}])], \tag{4}$$

where the correct last token prediction is $\boldsymbol{h}(a_{\bar{i}}^{\ell_{N_{\text{tr}}}})$. The following result shows that there exists an attention-only model that always gives correct predictions at any position.

**Proposition 3.** *Consider the input $\boldsymbol{X}$ in Eq. (4). There exists a one-layer attention-only model such that when $\|\boldsymbol{W}_{KQ}^h\| \to \infty$, it always gives the correct prediction for any token position.*

*Proof.* We present a construction for a 3-head model. For simplicity, we present the proof for the last subsequence, *i.e.*, $t \in \{\text{T} - S - 2, \ldots, \text{T}\}$, but it can be easily extended to other cases where $t < \text{T} - S - 2$ as well. Hereafter, we assume that $t \geq \text{T} - S - 2$.

At a high level, we use two heads, which we call the **relation** and **subject** heads, to get the output following $\boldsymbol{h}([\text{sep}])$, *i.e.*, $t = \text{T}$, and the third **grammar** head for other cases $t < \text{T}$. Let $\boldsymbol{p} := \sum_{i=1}^{S+2} \boldsymbol{h}(p_{-i})$, $\mathcal{T} := \{\text{T} - S, \ldots, \text{T} - 2\}$.

For the **relation** head, we set

$$\boldsymbol{W}_{KQ}^{\text{rel}} = \beta \left( \sum_\ell \boldsymbol{h}(r_\ell) + \boldsymbol{p} \right) \boldsymbol{h}([\text{sep}])^\top, \quad \boldsymbol{W}_{OV}^{\text{rel}} = \sum_\ell \sum_j \boldsymbol{h}(u_j^\ell) \boldsymbol{h}(r_\ell)^\top.$$

Then, when $\beta \to \infty$, the outputs of this head are as follows. When $t = \text{T}$, $g_{\text{rel}}(\boldsymbol{X}) = \boldsymbol{h}(r_{\ell_{N_{\text{tr}}}})$, *i.e.*, the most recent relation token in the sequence, and $f_{\text{rel}}(\boldsymbol{X}) = \sum_{j, \ell_{N_{\text{tr}}}} \boldsymbol{h}(u_j^{\ell_{N_{\text{tr}}}})$, *i.e.*, all attributes of type $\ell_{N_{\text{tr}}}$.

On the other hand, when $t < \text{T}$, $f_{\text{rel}}^t(\boldsymbol{X}) = \frac{1}{t} \sum_{j, \ell_{N_{\text{tr}}}} \boldsymbol{h}(u_j^{\ell_{N_{\text{tr}}}})$.

For the **subject** head, we set

$$\boldsymbol{W}_{KQ}^{\text{subj}} = \beta (\sum_j \boldsymbol{h}(s_j)) \boldsymbol{h}([\text{sep}])^\top, \quad \boldsymbol{W}_{OV}^{\text{subj}} = -\sum_j \left( \sum_{j' \neq j, \ell} \boldsymbol{h}(a_{j'}^\ell) \right) \boldsymbol{h}(s_j)^\top$$

Then, when $\beta \to \infty$, the outputs of this head are as follows. When $t = \text{T}$, $g_{\text{subj}}(\boldsymbol{X}) = \boldsymbol{h}(s_{\bar{i}})$, the subject token in the sequence, and $f_{\text{subj}}(\boldsymbol{X}) = -\left( \sum_u \boldsymbol{h}(u) - \sum_\ell \boldsymbol{h}(a_{\bar{i}}^\ell) \right)$, *i.e.*, negative of all attributes that are not associated with subject $s_{\bar{i}}$.

On the other hand, when $t < \text{T}$, $f_{\text{subj}}^t(\boldsymbol{X}) = -\frac{1}{t} \left( \sum_u \boldsymbol{h}(u) - \sum_\ell \boldsymbol{h}(a_{\bar{i}}^\ell) \right)$.

For the **grammar** head, we set

$$\boldsymbol{W}_{KQ}^{\text{grm}} = \beta \Big( \sum_{\ell, j} \boldsymbol{h}(u_j^\ell) \boldsymbol{h}(u_j^\ell)^\top + \sum_j \boldsymbol{h}(s_j) \boldsymbol{h}(s_j)^\top + \sum_\ell (\boldsymbol{h}(r_\ell) + \boldsymbol{h}([\text{sep}]) + \boldsymbol{p}) \boldsymbol{h}(r_\ell)^\top$$

$$+ \Big( \sum_\ell \boldsymbol{h}(r_\ell) + \boldsymbol{h}([\text{sep}]) + \boldsymbol{p} \Big) \sum_{\tilde{g}} \boldsymbol{h}(\tilde{g})^\top \Big)$$

$$\boldsymbol{W}_{OV}^{\text{grm}} = \sum_j \sum_\ell \boldsymbol{h}(s_j) \boldsymbol{h}(a_j^\ell)^\top + \Big( \sum_{\tilde{g}} \boldsymbol{h}(\tilde{g}) + \sum_\ell \boldsymbol{h}(r_\ell) \Big) \Big( \sum_j \boldsymbol{h}(s_j)^\top + \boldsymbol{h}([\text{sep}])^\top \Big)$$

$$+ \Big( 0.5 \sum_{\tilde{g}} \boldsymbol{h}(\tilde{g}) + \boldsymbol{h}([\text{sep}]) \Big) \sum_\ell \boldsymbol{h}(r_\ell)^\top.$$

Let $g_h^t = g_h^t(\boldsymbol{X})$ and similarly $f_h^t := f_h^t(\boldsymbol{X})$. The outputs in this case, when $\beta \to \infty$, are as follows:

- $t = \text{T} - S - 2$: $g_{\text{grm}} = \boldsymbol{h}(a_{\bar{i}}^{\ell_{N-1}})$, $f_{\text{grm}} = \boldsymbol{h}(s_{\bar{i}})$

- $t = \mathrm{T} - S - 1$: $g_{\mathrm{grm}} = \boldsymbol{h}(s_{\bar{i}})$, $f_{\mathrm{grm}} = \sum_{v \in \mathcal{G} \cup \mathcal{R}} \boldsymbol{h}(v)$

- $t \in \mathcal{T}$: $g_{\mathrm{grm}} = \begin{cases} 0.5(\boldsymbol{h}([\mathrm{sep}]) + \boldsymbol{h}(r_\ell)), & \text{if } \boldsymbol{x}_{t-S-2:t} \in \mathcal{R} \\ \boldsymbol{h}([\mathrm{sep}]), & \text{if } \boldsymbol{x}_{t-S-2:t} \notin \mathcal{R} \end{cases}$,

  $f_{\mathrm{grm}} = \begin{cases} \sum_{v \in \mathcal{G} \cup \mathcal{R}} \boldsymbol{h}(v), & \text{if } g_{\mathrm{grm}} = \boldsymbol{h}([\mathrm{sep}]) \\ 0.75 \sum_{v \in \mathcal{G}} \boldsymbol{h}(v) + 0.5 \sum_{v \in \mathcal{R}} \boldsymbol{h}(v) + 0.5\boldsymbol{h}([\mathrm{sep}]), & \text{if } g_{\mathrm{grm}} = 0.5(\boldsymbol{h}([\mathrm{sep}]) + \boldsymbol{h}(r_{\ell_{N_{\mathrm{tr}}}})) \end{cases}$

- $t = \mathrm{T} - 1$: $g_{\mathrm{grm}} = \boldsymbol{h}(r_{\ell_{N_{\mathrm{tr}}}})$, $f_{\mathrm{grm}} = \boldsymbol{h}([\mathrm{sep}]) + 0.5 \sum_{v \in \mathcal{G}} \boldsymbol{h}(\tilde{g})$

- $t = \mathrm{T}$: $g_{\mathrm{grm}} = \frac{1}{\mathrm{T}} \sum_t \boldsymbol{x}_t$, $f_{\mathrm{grm}} = \frac{N_{\mathrm{tr}}-1}{\mathrm{T}} \boldsymbol{h}(s_{\bar{i}}) + \frac{N_{\mathrm{tr}}}{\mathrm{T}} \left( 1.5 \sum_{v \in \mathcal{G}} \boldsymbol{h}(v) + \sum_\ell \boldsymbol{h}(r_\ell) + \boldsymbol{h}([\mathrm{sep}]) \right)$

Combining the outputs of the individual heads, the final output in different cases is as follows.

$$
v^* = \begin{cases} a_{\bar{i}}^{\ell_{N_{\mathrm{tr}}}}, & \text{if } t = \mathrm{T}, \\ [\mathrm{sep}], & \text{if } t = \mathrm{T} - 1, \\ v \sim \mathrm{Unif}(\mathcal{G} \cup \mathcal{R}), & \text{if } t \in \mathcal{T}, \nexists j \in \{t - S - 2, ..., t\}, \boldsymbol{x}_j \in \mathcal{R} \\ v \sim \mathrm{Unif}(\mathcal{G}), & \text{if } t \in \mathcal{T}, \exists j \in \{t - S - 2, ..., t\}, \boldsymbol{x}_j \in \mathcal{R} \\ v \sim \mathrm{Unif}(\mathcal{G} \cup \mathcal{R}), & \text{if } t = \mathrm{T} - S - 1, \\ s_{\bar{i}}, & \text{if } t = \mathrm{T} - S - 2. \end{cases}
$$

$\square$

Next, consider `ICL` sequences of the form

$$
\boldsymbol{X} = [\boldsymbol{h}(s_{j_1}), \boldsymbol{h}([\mathrm{sep}]), \boldsymbol{h}(a_{j_1}^{\bar{\ell}}), ..., \boldsymbol{h}(s_{j_{N_{\mathrm{ft}}+1}}), \boldsymbol{h}([\mathrm{sep}])], \tag{5}
$$

where the correct last token prediction is $\boldsymbol{h}(a_{j_{N_{\mathrm{ft}}}}^{\bar{\ell}})$. The following result shows that there exists an attention-only model that always gives correct predictions for these sequences.

**Proposition 4.** *Consider the input $\boldsymbol{X}$ in Eq. (5). There exists a one-layer attention-only model such that when $\|\boldsymbol{W}_{KQ}^h\| \to \infty$, it always gives the correct prediction for the last token position.*

*Proof.* We present a construction for a 3-head model, with minimal changes to the construction in the proof of Proposition 1. Specifically, the **grammar** head is unchanged. At a high level, the **subject** head attends to the last/query subject and maps it to the negative of the sum of attributes not associated with it, while the **relation** head attends to the attributes in the context, and maps them to all attributes of the same type $\bar{\ell}$.

For the **subject** head, we set

$$
\boldsymbol{W}_{KQ}^{\mathrm{subj}} = \beta \Big( \sum_j \boldsymbol{h}(s_j) + \boldsymbol{h}(p_{-1}) \Big) \boldsymbol{h}([\mathrm{sep}])^\top, \quad \boldsymbol{W}_{OV}^{\mathrm{subj}} = -\sum_j \left( \sum_{j' \neq j, \ell} \boldsymbol{h}(a_{j'}^\ell) \right) \boldsymbol{h}(s_j)^\top
$$

Then, when $\beta \to \infty$, $g_{\mathrm{subj}}(\boldsymbol{X}) = \boldsymbol{h}(s_{j_{N_{\mathrm{ft}}}})$, *i.e.*, the query subject, and $f_{\mathrm{subj}}(\boldsymbol{X}) = -\Big( \sum_u \boldsymbol{h}(u) - \sum_\ell \boldsymbol{h}(a_{j_{N_{\mathrm{ft}}}}^\ell) \Big)$.

Let $\mathcal{S}' \subset \mathcal{S}$ denote a subset of subjects. We define the set of unique attributes for the subjects in $\mathcal{S}'$: for each attribute type $\ell$, let $\mathcal{U}_\ell' := \cup_{j \in \mathcal{S}'} \{a_j^\ell\}$. For the **relation** head, we set

$$
\boldsymbol{W}_{KQ}^{\mathrm{rel}} = \beta \sum_\ell \sum_{u \in \mathcal{U}_\ell'} \boldsymbol{h}(u) \boldsymbol{h}([\mathrm{sep}])^\top, \quad \boldsymbol{W}_{OV}^{\mathrm{rel}} = \sum_\ell \Big( \sum_j \boldsymbol{h}(u_j^\ell) \Big) \Big( \sum_{u \in \mathcal{U}_\ell'} \boldsymbol{h}(u) + \boldsymbol{h}(r_\ell) \Big)^\top.
$$

Then, when $\beta \to \infty$, $g_{\mathrm{rel}}(\boldsymbol{X}) = \frac{1}{N_{\mathrm{ft}}} \sum_i \boldsymbol{h}(a_{j_i}^{\bar{\ell}})$, *i.e.*, the average of the attributes that appear in the context, and $f_{\mathrm{rel}}(\boldsymbol{X}) = \sum_j \boldsymbol{h}(u_j^{\bar{\ell}})$.

Combining the outputs of the individual heads, the model output is

$$f(\boldsymbol{X}) = \sum_j \boldsymbol{h}(u_j^{\bar{\ell}}) - \left( \sum_u \boldsymbol{h}(u) - \sum_\ell \boldsymbol{h}(a_{j_{N_{\mathrm{ft}}}}^\ell) \right) + \frac{1}{\mathrm{T}} \sum_{i \in \mathcal{M}''} \boldsymbol{h}(s_i) + \frac{N_{\mathrm{ft}}}{\mathrm{T}} \sum_{v \in \mathcal{G} \cup \mathcal{R}} \boldsymbol{h}(v),$$

where $\mathcal{M}'' \subseteq [M]$. Then, the final prediction $v^* = a_{j_{N_{\mathrm{ft}}}}^{\bar{\ell}}$.

$\square$

## E.2 EXPERIMENTAL VALIDATION

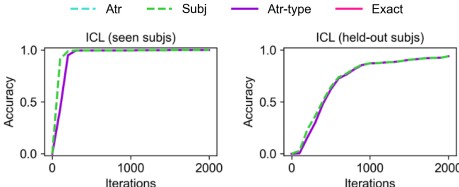

Figure 16: Finetuning the single-layer attention-only model pretrained on `PT` sequences using `ICL` sequences with a subset of subjects enables generalization on `ICL` sequences with held-out query subjects.

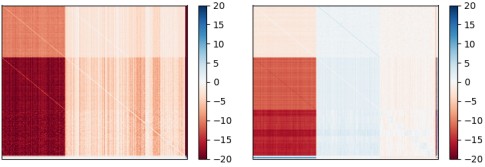

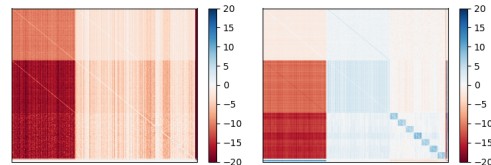

Figure 17: Visualization of the weight matrices $\boldsymbol{W}_{OV}^{\mathrm{subj}}$ (left) and $\boldsymbol{W}_{OV}^{\mathrm{rel}}$ (right) after pretraining the single-layer attention-only model on `PT` sequences.

Figure 18: Visualization of the weight matrices $\boldsymbol{W}_{OV}^{\mathrm{subj}}$ (left) and $\boldsymbol{W}_{OV}^{\mathrm{rel}}$ (right) after fine-tuning the single-layer attention-only model on `ICL` sequences.