# OpenReview forum: "Understanding Contextual Recall in Transformers: How Finetuning Enables In-Context Reasoning over Pretraining Knowledge"
_ICLR.cc/2026/Workshop/Sci4DL — Sci4DL 2026_

### Official Review · Reviewer_h4i7 · 2026-02-25

**Fit:** 2
**Significance:** 2
**Confidence:** 2

**Summary:**

This paper investigates "contextual recall," a specialized form of in-context learning where models must retrieve factual knowledge by implicitly inferring attribute types from provided examples, which requires both the acquisition of factual knowledge and adaptability to novel prompt formats. Through a controlled synthetic framework, the authors demonstrate that while pretraining facilitates knowledge acquisition, specific finetuning is essential to develop the inference mechanisms and low-dimensional representations required for zero-shot task recognition. This helps us to understand the different roles of pretraining and finetuning for in-context reasoning ability.

**Strengths:**

1. The introduced task of "contextual recall" itself is pretty novel, which effectively bridges the gap between static knowledge storage and dynamic in-context reasoning
2. The authors provide a solid synthetic framework and environment for task generation and learning, as well as analytical constructions for the mechanism understanding and insight discovery.

**Suggestions:**

1. It seems like the mechanistic analysis focuses on an attention-only model. But transformers also have other components, e.g., MLP. clarifying how MLPs interact with the "subject/relation" heads during recall would add depth
2. Currently, the attribute types are pretty small. Would the task vector clustering remain stable as the number of clusters increases toward more realistic scales?
3. The distribution data is generated based on Markov chain diversity. What about some other types of data distributions?

---

### Official Review · Reviewer_z9Py · 2026-02-27

**Fit:** 2
**Significance:** 2
**Confidence:** 3

**Summary:**

This paper studies "contextual recall," where models should infer an implicit attribute type from in-context examples to attributes of the last query. Using a controlled synthetic framework, the authors show pretraining alone is insufficient, but finetuning on a subset of subjects (even with a different prompt format) enables generalization to held-out subjects. Representational analysis reveals finetuning induces clustered latent encodings of attribute type.

**Strengths:**

- The synthetic setup is clean and well-designed, offering good experimental control.
- The findings are clearly presented and build logically on each other.

**Suggestions:**

- The rigid, PT format makes the failure of ICL to emerge from pretraining relatively unsurprising, as prior work (Raventos et al., 2023; Xie et al., 2022) has established that ICL tends to require diverse pretraining distributions.
- The representational analysis showing clustered task representations from in-context demonstrations is heavily reminiscent of work on induction heads and task vectors (Olsson et al., 2022; Todd et al., 2023), which are not cited or discussed.

---

### Meta-Review · Area_Chair_HnNS · 2026-03-01

**Recommendation:** Accept

**Metareview:**

Strong fit, fair contribution. Recommending acceptance.

---

### Decision · Program_Chairs · 2026-03-02

Accept